# A Novel Green Diluent for the Preparation of Poly(4-methyl-1-pentene) Membranes via a Thermally-Induced Phase Separation Method

**DOI:** 10.3390/membranes11080622

**Published:** 2021-08-13

**Authors:** Yuanhui Tang, Mufei Li, Yakai Lin, Lin Wang, Fangyu Wu, Xiaolin Wang

**Affiliations:** 1College of Chemistry and Environmental Engineering, China University of Mining and Technology, Beijing 100083, China; tyh@cumtb.edu.cn (Y.T.); limufei11@126.com (M.L.); 2Beijing Key Laboratory of Membrane Materials and Engineering, State Key Laboratory of Chemical Engineering, Department of Chemical Engineering, Tsinghua University, Beijing 100084, China; wanglin891208@mail.tsinghua.edu.cn (L.W.); wfy17@mails.tsinghua.edu.cn (F.W.)

**Keywords:** poly(4-methyl-1-pentene) membrane, thermally induced phase separation, green diluent, liquid-liquid phase separation, bicontinuous structure, myristic acid

## Abstract

The use of green solvents satisfies safer chemical engineering practices and environmental security. Herein, myristic acid (MA)—a green diluent—was selected to prepare poly- (4-methyl-1-pentene) (PMP) membranes with bicontinuous porous structure via a thermally induced phase separation (TIPS) process to maintain a high gas permeability. Firstly, based on the Hansen solubility parameter ‘distance’, Ra, the effect of four natural fatty acids on the PMP membrane structure was compared and studied to determine the optimal green diluent, MA. The thermodynamic phase diagram of the PMP-MA system was calculated and presented to show that a liquid-liquid phase separation region could be found during the TIPS process and the monotectic point was around 34.89 wt%. Then, the effect of the PMP concentration on the morphologies and crystallization behavior was systematically investigated to determine a proper PMP concentration for the membrane preparation. Finally, PMP hollow fiber (HF) membranes were fabricated with a PMP concentration of 30 wt% for the membrane performance characterization. The resultant PMP HF membranes possessed good performances that the porosity was 70%, the tensile strength was 96 cN, and the nitrogen flux was 8.20 ± 0.10 mL·(bar·cm^2^·min)^−1^. We believe that this work can be a beneficial reference for people interested in the preparation of PMP membranes for medical applications.

## 1. Introduction

Poly(4-methyl-1-pentene) (PMP) is a semi-crystalline polyolefin that has several interesting properties, such as high thermal stability, good chemical resistance, excellent gas permeability, and biosecurity [1]. It is of practical importance as a membrane-making polymer, and the PMP membranes are found to serve in many practical applications, such as gas separation [2] and some essential membrane materials applied in medical science, especially in extracorporeal membrane oxygenation (ECMO) systems [3,4], which are extracorporeal devices providing prolonged cardiac and respiratory support to persons whose heart and lungs are unable to provide an adequate amount of gas exchange or perfusion to sustain life [5]. At present, the PMP membranes applied in the ECMO systems are mainly prepared via thermally induced phase separation (TIPS) [6,7], since the TIPS process has better controllability in the membrane preparation and excellent membrane performances such as high porosity and good mechanical property can be easily attained.

From 1990 to 2020, dozens of studies were conducted to prepare PMP membranes via the TIPS method and optimize the preparation conditions, the key results of which are summarized in Table 1. 

According to these studies, it can be known that a suitable diluent was the most decisive factor for controlling the structure of the PMP membranes via the TIPS process. This is because that as the core materials to realize the gas exchange in the ECMO systems, the PMP membranes should be hydrophobic, and possess structures that consist of a uniform and bicontinuous porous cross-section, and a thin dense surface layer to ensur a high gas permeability and satisfy blood rejection [23,24]. Commonly, as the interaction between the polymer and the diluents is comparatively good, a solid-liquid (S-L) phase separation happens, resulting in crystalline structures with lower porosity, while as the interaction between the polymer and the diluents is comparatively weak, a liquid-liquid (L-L) phase separation may occur, resulting in a bicontinuous or cellular structure with a higher porosity [25,26]. Therefore as listed in Table 1, different organic solvents including dioctyl adipate (DOA), diphenyl ether (DPE), dibutyl phthalate (DBP), dioctyl phthalate (DOP) and mineral oil (MO), have all been chosen as the single diluent to prepare the PMP membranes for medical applications. Only the S-L phase separation occurred in the MO or DOA system and membranes with flaky crystal structures were normally obtained [6,13,14]; whereas the L-L phase separation could be found in the DOP, DPE, or DBP system and then the membranes with the bicontinuous structure would be obtained [7,15,16,17]. Also, it can be found that among the various diluent systems, only several single diluents were adopted while most of the systems used the diluent mixture to regulate the membrane structure, which is because that it is very hard to find an appropriate single diluent to prepare a PMP membrane with desirable structure via TIPS, although production and processing of the membrane with a single diluent are relatively simpler. Besides, considering the actual medical application, the PMP hollow fiber (HF) membranes were mainly prepared for the membrane performance tests. In order to improve the membrane performance, recently people have been paying more attention to studying and understanding the effects of different process factors, including the polymer chain conformation [18], polymer concentration [19], cooling rate [21,23], diluent extractant [18], coagulation medium [9,11], et al., on the membrane structures and performances including porosity [7], dense layer [17], and mechanical properties [8] in detail. A higher gas permeability, higher water resistance, and good mechanical strength were considered as the ultimate goal for these studies.

Although PMP membranes were primarily prepared for medical applications, the detrimental impact of these diluents on the environment and human health has not been considered in preparing membranes. Most of the diluents listed in Table 1 such as DOA, DOP, and DBP, were a little toxic and some were even known to be carcinogenic [27], which might lead to the resultant PMP membranes having a negative impact on the health of humans and the environment. Thus, in response to the national call of “green and pollution-free”, finding a green and safe diluent is an important topic in preparing the PMP membranes for the ECMO application [28,29]. In chemistry, particularly in biochemistry, a fatty acid is a carboxylic acid with a long aliphatic chain, which is either saturated or unsaturated [30]. Fatty acids are a major component of the lipids (up to 70 wt%) in some species such as microalgae [31] although in some other organisms are not found in their standalone form, instead exist as three main classes of esters: triglycerides, phospholipids, and cholesteryl esters. In any of these forms, fatty acids are both important dietary sources of fuel for animals and important structural components for cells. Therefore, fatty acids obviously belong to non-toxic green solvents which have no harmful impact on the environment and human health [32].

In this manuscript, four nontoxic fatty acids that are extensively existed in nature, including lauric acid (LA), myristic acid (MA), palmitic acid (PA), and stearic acid (SA), were compared and studied to reveal their possibilities to be the diluents of PMP. Then considering the MA as the target green diluent, the phase diagram, the effect of the PMP concentration on the membrane structure as well as crystallization behavior were investigated to explore a suitable PMP concentration for the membrane preparation and performance characterization. Then, the PMP HF membranes were prepared with MA as the diluent and a PMP concentration of 30 wt% to study the membrane performances including porosity, tensile strength, and nitrogen flux. The nontoxic green diluent is conducive to the safety of membrane fabrication and subsequent medical applications.

## 2. Experimental

### 2.1. Materials and Chemicals

PMP (*Mn* = 87,200 g·mol^−1^, MFR = 26 g/10min, P = 5 kg, T = 260 °C) was provided by Mitsui Chemical (Tokyo, Japan). PA (Purity: AR) and SA (Purity: AR) were purchased from Meryer (Shanghai, China) Chemical Technology Co., Ltd. (Shanghai, China). LA (Purity: AR) was purchased from Shanghai Chemical Technology Co., Ltd. (Shanghai, China). (MA (Purity: AR) was purchased from Beijing Jinming Biological Technology Co., Ltd. (Beijing, China). Anhydrous ethanol was purchased from Shanghai Titan Technology Co., Ltd. (Shanghai, China), and used as the extractant. Isobutanol was purchased from Shanghai Titan Technology Co., Ltd. (Shanghai, China) and used as a liquid for the porosity measurement. All chemicals used in this study were not purified further. The chemical formulas, the molecular structures, and the basic physical properties of the PMP and fatty acids adopted in the work were listed in Table 2.

### 2.2. Preparation of the PMP Membranes

#### 2.2.1. Preparation of the PMP-MA Mixture Samples

PMP and the MA were melt-mixed by using a mini twin-screw extruder (ULTnano TW05, Technovel Corporation, Osaka, Japan) and then quenched into a cooling liquid nitrogen bath to obtain the PMP-MA mixture samples. The rate and temperature of the extruder were set at 300 rpm and 240 °C, respectively.

#### 2.2.2. Preparation of the PMP-Diluents Piece Membrane Samples

PMP and the diluents were melt-mixed by using a mini twin-screw extruder (ULTnano TW05, Technovel Corporation, Osaka, Japan) and then quenched into a cooling water bath. The membrane precursors were extracted with anhydrous ethanol to remove the diluents and obtain the piece membrane samples. The rate and temperature of the extruder were set at 300 rpm and 240 °C, respectively.

#### 2.2.3. Preparation of the PMP HF Membranes

The PMP HF membranes were prepared via the TIPS method with a PMP concentration of 30 wt% by using a twin-screw extruder (KZW15TW, Technovel Corporation, Osaka, Japan), as shown in Figure 1. The casting solution formed from the PMP and MA was extruded from the spinneret, quenched into a cooling bath, and wound by a take-up roller. Bore fluid was introduced to the inner orifice to make the lumen of HFs. The cooling medium and bore fluids were water and glycerol, respectively. The temperatures of the spinneret, cooling bath, and bore fluid was set to be 240, 40, and 180 °C, respectively. The dope flow rate was 16.70 g/min^−1^. The air gap between the spinneret and the cooling bath was 40 mm. The fibers were extracted with anhydrous ethanol and the dry PMP HF membranes were obtained after the volatilization of ethanol.

### 2.3. Phase Diagram and Crystallization Kinetics

The cloud points of these PMP-MA mixture samples were detected visually by the appearance of turbidity under an optical microscope (Olympus BX51, Tokyo, Japan). The small pieces of the samples were placed between a pair of microscope coverslips. A hot stage (THMS600, Linkam, Epsom, UK) was adopted to heat the samples from the room temperature to 260 °C and then hold for 5 min and then cool to 50 °C at a rate of 10 °C/min.

The dynamic crystallization temperatures were measured by a differential scanning calorimetry (DSC, TA Q200, TA Instruments, New Castle, DE, USA). The PMP-MA mixture samples in an aluminum pan were kept at 260 °C for 5 min and cooled to 50 °C at a rate of 10 °C/min. The exothermic peak in the cooling was taken as the dynamic crystallization temperature of the sample.

The melting data were used to draw the DSC curves and calculate the PMP crystallinity $X_{c}$ by the following equation:(1)$$X_{c}=\frac{H_{S}}{X_{100}\times C}\times 100\%$$
where $X_{100}$ = 117.20 J/g was the melting enthalpy for a 100% crystalline sample of PMP, and $H_{S}$ was the melting enthalpy of the PMP-MA mixture samples measured by the DSC, C was the polymer concentration in unit 1.

### 2.4. Characterization of PMP Membranes

#### 2.4.1. Crystalline Forms

X-ray diffractometer (XRD) analysis was conducted to determine phase identification and crystalline nature of prepared membranes [33]. The crystallization forms of the PMP-MA piece membrane samples were characterized by an XRD (D8, Bruker, Mannheim, Germany). The working electric current and voltage were set as 40 mA and 40 kV, respectively. The Cu: Kα radiation was employed on the samples. The scanning angle 2*θ* was from 5° to 50° with a step size of 0.02° and a time step of 0.10 s.

#### 2.4.2. Morphology Study

The morphology of the PMP membrane was examined using a scanning electron microscope (SEM, JEOL JSM-7401, Tokyo, Japan). It also gives information about structure i.e., porous or dense membrane [34]. The piece membrane samples were fractured in liquid nitrogen and coated with platinum. The SEM with the accelerating voltage of 3 kV was used to examine the cross-section and surface morphologies of the membranes.

The surface roughness of the PMP HF membranes was measured by an atomic force microscope (AFM, MultiMode 8HR, Bruker, Billerica, MA, USA).

#### 2.4.3. Porosity

After measuring the dry weight $M_{d}$ of the PMP HF membranes, they were immersed in the isobutanol for 48 h to become wet. And then wiped off the isobutanol on the membrane surface, the wet weight $M_{w}$ of the PMP HF membranes was measured. The porosity of the PMP HF membranes was calculated according to the following equation:(2)$$\varepsilon =\frac{\frac{M_{w}-M_{d}}{\rho_{k}}}{\frac{M_{w}-M_{d}}{\rho_{k}}+\frac{M_{d}}{\rho_{p}}}$$
where $M_{w}$ and $M_{d}$ were the weights of the wet and dry membranes, respectively, $\rho_{k}$ and $\rho_{p}$ were the densities of the isobutanol and PMP (0.81 g/cm^3^ and 0.83 g/cm^3^), respectively.

#### 2.4.4. Surface Contact Angle

The surface contact angle of water was defined as the angle between the tangent line of the membrane surface and the contact point of a water droplet on the membrane surface. Surface contact angle measurement at room temperature was made using a contact angle meter (OCA20, Dataphysics, Filderstadt, Germany). A one μL water droplet was placed on the membrane surface to capture the surface contact angle. Each membrane was measured at different locations at least five times and the average value was reported.

#### 2.4.5. Nitrogen Flux of the Membrane

At room temperature, nitrogen was injected into the PMP HF membrane with a certain length at different pressures (0.10–0.90 bar), and the nitrogen flux F was measured with a soap film flowmeter. The nitrogen flux calculation formula was as follows:(3)$$F=\frac{V}{P\times A\times t}$$
where V was the volume of a bubble flowing in the soap film flowmeter (unit is mL); and P was the pressure in the flow process, in-unit bar; A was the effective area of the PMP HF membrane, in cm^2^; t was the time for a bubble to flow through a certain volume of the soap film flowmeter, in the unit of a minute.

#### 2.4.6. Mechanical Property

The mechanical property of the PMP HF membranes was evaluated using a universal testing machine (AGS-J 200 N, Shimadzu, Kyoto, Japan) at room temperature. The PMP HF membranes were vertically fixed between two pairs of tweezers at a length of 50 mm and extended at a constant elongation rate of 50 mm/min until the membranes were broken. Tensile strength was calculated using the unit cross-sectional areas of the HF membranes. At least five HF membranes were used to get a mean mechanical property.

## 3. Result and Discussion

### 3.1. Diluent Selection

This section studied the effect of different fatty acids on the cross-section structure of the PMP membranes prepared via TIPS to initially help to select a proper diluent. Figure 2 showed the cross-section membrane structures of the PMP piece membranes prepared by different fatty acids as the PMP concentration was set to be 30 wt%. As shown in Figure 2, the cross-section structure of the LA system was a typical flaky crystal, which was normally resulted from the S-L phase separation. This kind of structure always brings about a lower gas permeability that is not desirable. When the diluent was MA, PA, and SA, the membrane structure gradually changed to bicontinuous and cellular, which indicated the L-L phase separation should occur during the quenching process of the three diluent systems. As pointed by the literature [15,17], this phenomenon also signified that the compatibility between PMP and these fatty acids gradually became weaker in the order of LA, MA, PA, and SA. In addition, concerning the difference in the membrane structure, the compatibility between PMP and the three fatty acids should be distinguished from each other.

The interaction between the polymer and solvents could be quantitatively characterized by the Hansen solubility parameter δ (HSP), which is normally divided into three partial components: the dispersion part $\delta_{D}$, the polar part $\delta_{P}$, the hydrogen bonding part $\delta_{H}$ [35]. Table 3 summarized the three HSP components, ‘Ra’ parameters, and the corresponding membrane structure of the PMP-diluents system based on the group contribution method [36,37] and the manual of Hansen solubility parameters [38,39]. Herein, the Ra parameter is normally called the solubility parameter ‘distance’ and calculated between two materials based on their respective partial solubility parameter components. It represents the compatibility of the two components and a higher Ra parameter often indicates weaker compatibility. The detailed information and calculation method about the Ra and the HSPs of the different components were listed in Appendix A. As listed in Table 3, the Ra parameter was increased from 2.60 to 3.70 in the order of LA, MA, PA, and SA, which meant the interaction between PMP and these fatty acids was getting worse, and the phenomena also agreed with the results of the evolution of the membrane structure: changing from flaky crystal to bicontinuous and cellular. Whereas it should be noted that the cross-section of the membranes that presents bicontinuous or cellular is more determined by the phase diagram of the system and the process condition, rather than the compatibility of the PMP-diluent system. As assistant evidence, Table 3 also listed the corresponding parameters and experimental results of the PMP-diluents systems reported by the related references, including DBP, DPE, DOP, and DOA. The results also showed that the Ra can help to predict the membrane structure change, which was also consistent with the results of this study. In addition, considering that the bicontinuous structure brought about higher porosity and a larger gas flux than the cellular or flaky crystal structure. In the following part, the MA was regarded as the target green diluent to prepare the PMP membranes.

### 3.2. Phase Diagram of the PMP-MA System

Thermodynamic phase diagrams had traditionally been the primary source of guidance for understanding the outcome of the TIPS process. Figure 3a displays the thermodynamic phase diagram of the PMP-MA system and Figure 3b presents the linear relationship between the interaction parameter χ of the PMP-MA system and the reciprocal of temperature. The calculation method of the interaction parameter χ and the phase diagram was listed in Appendix B. A linear relationship of Figure 3b signified that the phase diagram of Figure 3a was relatively reliable [40]. The phase diagram showed that the system had a monotectic point ($\Phi_{m}$) of approximately 34.89 wt%, indicating that when the PMP concentration was below 34.89 wt%, the L-L phase separation occurred before the polymer crystallization because the binodal curve was higher than the crystallization temperature curve. The L-L phase separation regions were mainly divided into the metastable region (the gap between the binodal curve and the spinodal curve) and the unstable region (below the spinodal curve). Based on the thermodynamics of the polymer-diluent system, if the system dropped into the metastable region during the quenching process, the cellular structure was easily formed due to nucleation-growth kinetics, and if the system dropped into the unstable region during the quenching process, the bicontinuous structure was normally formed due to spinodal decomposition kinetics [41,42]. Thus, the bicontinuous or cellular structure formation was expected when the PMP concentration was below 34.89 wt%.

### 3.3. The Effect of the PMP Concentration on the Crystallization Behavior and Morphologies of the Membranes

It has been demonstrated that the polymer concentration has a critical effect on the crystalline forms and morphology of the polymeric membranes prepared via the TIPS process [26]. As a result, this section mainly explored the effect of the PMP concentration on the membrane structure of the PMP-MA system to determine a suitable concentration for the following PMP membrane preparation and application. Table 4 and Figure 4 show the data of peak points, crystallinities, and the DSC curves of the PMP-MA system with different PMP concentrations. 

Most of the data listed in Table 4 has been added into Figure 4 to make them more comprehensive. During the crystallization process, only one endothermic peak appeared. In addition, the peak shifted towards higher temperatures, and the melting enthalpies rose with the increase of the PMP concentration. This is because that more polymer chains would ask for more energy to be melt. Dual-melting peaks appeared in the melting process of the system. With the increase of the PMP concentration, the melting enthalpies gradually increased, and the dual-melting peaks moved towards higher temperatures and then overlapped until the PMP concentration was 100 wt%. The existence of dual-melting peaks in the melting process might be caused by partial melting of the original single crystals at lower temperatures (primary crystallization) followed by melting of the crystals with thicker lamellae by partial melting and recrystallization at higher temperatures (secondary crystallization) [14,15,43]. Thus, the reason why the melting enthalpies rose and the dual-melting peak finally overlapped with the increase of the PMP concentration could be attributed to the addition of diluent (MA). Because considering a comparatively strong interaction between the PMP and MA, as the composition of MA was decreased, the viscosity of the PMP casting solution should be increased obviously [16].

Figure 5a shows the X-ray diffractograms of the PMP-MA system prepared with different PMP concentrations, and Figure 5b presents a schematic diagram of the top and side views of the carbon chain conformation in the crystalline phase of PMP. According to the literature [44,45], there were at least five different crystalline forms of PMP: from I to V. The X-ray diffractograms of all the sample surfaces of Figure 5a were seen to have well-defined peaks at 2*θ* about 9.50°, 13.40°, 16.70°, 18.30°, 20.70°, and 21.60°, which corresponded to (200), (220), (212), (321), (113/411), and (203/322) crystal plane of the I-phase [46,47]. In addition, with the PMP concentration increased from 10 wt% to 40 wt%, the X-ray diffractograms were hardly influenced. Thus, by combining Figure 4b and Figure 5a, it can be speculated that no matter how the position of the dual-melting peak moved, the two crystals were of the same crystalline type. Medellin-Rodriguez attributed this phenomenon to the secondary crystallization which would not give rise to the diffraction, just giving place to perceptive X-ray scattering [48,49]. Therefore, the difference of the secondary crystallization resulting from different PMP concentrations affected the DSC melting curves and the overall crystallinities, but not the diffractograms. Besides, the polymer chains of the I crystalline phase were characterized by a 7_2_ helical conformation and a chain packing in a tetragonal unit cell with a = 18.66Å and c = 13.80 Å (as shown in Figure 5b), and could be easily obtained from the melt or from crystallization in high boiling solvents, which was the most stable crystalline form [50]. Because of the rise in the viscosity of the system, the total crystallinities of Table 4 gradually increased as the PMP concentration increased via S-L phase separation; A higher crystallinity occasionally appeared in Table 4 when the PMP concentration was lower than 34.89 wt%, such as 20 wt%, which might denote that the mass exchange rate in dilute casting solutions when the PMP concentration was a little lower was higher than that in the higher concentration case, allowing the polymer chains could have more time and space to attain higher crystallinities [51].

The cross-section structures of the piece membranes that were obtained with different PMP concentrations are shown in Figure 6. According to the phase diagram of Figure 3a, since the PMP concentrations of 10 wt% and 20 wt% were located in the unstable region, the cross-section membrane structures presented a bicontinuous, and as the PMP concentration was increased to 30 wt%, a cellular structure could be found. Furthermore, when the PMP concentration was higher than 34.89 wt% (the monotectic point), the whole system should all undergo the S-L phase separation process to form a flaky crystal and change to a fuzzy crystal structure. The evolution of the membrane morphologies of Figure 6 indeed agreed with the phase diagram of Figure 3a. Figure 6 also shows that as the PMP concentration was increased, the porosity decreased gradually. Based on the previous research, the mechanical property was usually improved when the PMP concentration was higher [52]. Concerning that the membrane could possess high porosity and good mechanical property at the same time, 30 wt% was regarded as an appropriate concentration for the following PMP HF membrane preparation.

### 3.4. Morphology and Performance of the PMP HF Membranes

In this section, we wanted to characterize the performance of the PMP membranes. According to Table 1, the hollow fiber (HF) geometry was commonly adopted for ECMO systems [22,24]. Besides, the PMP piece membranes prepared in Section 3.3 were a little small and hard to obtain stable performance data. As a result, the PMP HF membranes were prepared for the performance characterization when the MA was chosen as the diluent and the PMP concentration was set to be 30 wt%.

#### 3.4.1. Morphology of the PMP HF Membranes

The structure of the PMP HF membrane of the PMP-MA system is shown in Figure 7 as the PMP concentration was set to be 30 wt%. Figure 7b shows that there was an ultrathin and dense surface layer existing on the porous sublayer and the dense surface layer was estimated to be 0.21 μm, which would be conducive to keep a high gas permeability and a good liquid resistance; Figure 7a,c display the overall and partial cross-section structures of the membrane, and the membrane cross-section presented a uniform bicontinuous porous structure, which was resulted from the occurrence of the L-L separation during the quenching process, and also beneficial to obtain a higher gas permeability; Figure 7d obviously shows that the outer surface of the membrane was dense and nonporous, which could effectively reject liquid and big molecules. Figure 7e shows the roughness of the membrane surface which was obtained by the AFM testing. The arithmetic means roughness (Ra) and root-mean-square roughness (Rq) of the PMP HF membranes were 33.10 nm and 42.40 nm, respectively. In summary, the morphology of the PMP HF membranes possessed the bicontinuous cross-section structure beneath an ultrathin dense surface layer, which was significant for gas permeability applied in the ECMO system.

#### 3.4.2. Performance of the PMP HF Membranes

Table 5 lists the configuration information and gas permeability performance of the PMP HF membranes. As reference evidence, Table 5 also summarizes some of the corresponding literature data from Table 1. As a whole, the PMP HF membrane was very thin due to the fact the outer and inner diameters of the fiber were only around 900 and 600 μm, which meant that the thickness of the fiber was only around 150 μm. The data was comparable to the literature results, which signified that the surface dense layer was thin enough to keep a high gas permeability. Also owing to the bicontinuous cross-section structure of the membrane, the porosity was as high as 70%, and the nitrogen flux was 8.20 ± 0.10 mL·(bar·cm^2^·min)^−1^. The surface contact angle was 105.8°, indicating that the PMP HF membrane can be regarded as hydrophobic membranes, and this should be attributed to the existence of methyl branches of the PMP molecules that reduced the ability to form hydrogen bonds in water. The tensile strength was 96 cN, which was only at an intermediate level of the reference data and may need further improvement. In general, Table 5 proves that the performances of the PMP HF membranes of this work basically exceeded or met the existing reference data. Therefore, MA was considered as a desirable green diluent to prepare the PMP HF membranes, and the resultant membranes could have ideal structures and gas permeability simultaneously. Although more studies need to be conducted to further promote the mechanical strength as well as improve the gas permeability.

## 4. Conclusions

A green diluent—MA—was successfully found to be a proper single diluent to prepare PMP HF membranes for the ECMO application via the TIPS process. Firstly, by combining the Hansen solubility parameter and the resultant membrane structure, MA was selected to be the optimal diluent among a series of natural green fatty acids. The phase diagram of the PMP-MA system showed that the monotectic point was 34.89 wt%, which meant with the increase of the PMP concentration, the phase separation process was gradually changed from the L-L to the S-L phase separation, and the membrane structure would change from bicontinuous to a cellular and crystal structure. Then, the effect of the PMP concentration on the morphologies and crystallization behavior was systematically investigated to determine that 30 wt% was an appropriate choice for the PMP membrane preparation. Because for the PMP-MA system, when the PMP concentration is 30 wt%, the bicontinuous cross-section could be obtained due to the L-L phase separation, and the mechanical strength can also be guaranteed since a higher polymer concentration results in a better mechanical strength. In consideration of the membrane performance tests and future applications, the PMP HF membranes were fabricated with a PMP concentration of 30 wt%. The resultant membranes presented that the porosity was 70%, the tensile strength was 96 cN, and the nitrogen flux was 8.20 ± 0.10 mL·(bar·cm^2^·min)^−1^, which are better or comparable to the results seen in previous research. The experimental data proved that MA can be regarded as a satisfactory diluent to prepare a PMP membrane with a bicontinuous cross-section structure beneath an ultra-thin dense surface layer. This work provided a successful way to select a green diluent—MA—and PMP HF membranes with desirable performance were also prepared. In the future, more detailed studies will be systematically conducted to optimize the system constituents, process parameters, and surface modification to realize advanced preparation of the hollow fiber PMP membranes that can be successfully applied in the medical field, especially for the ECMO application.

## Figures and Tables

**Figure 1 membranes-11-00622-f001:**
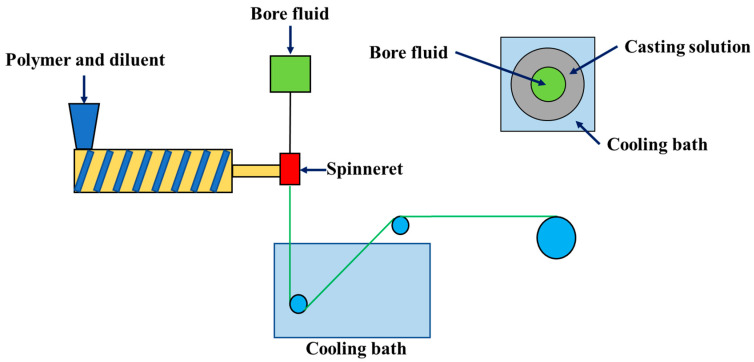
The schematic diagram of the preparation of the PMP HF membranes via the screw extruder system.

**Figure 2 membranes-11-00622-f002:**
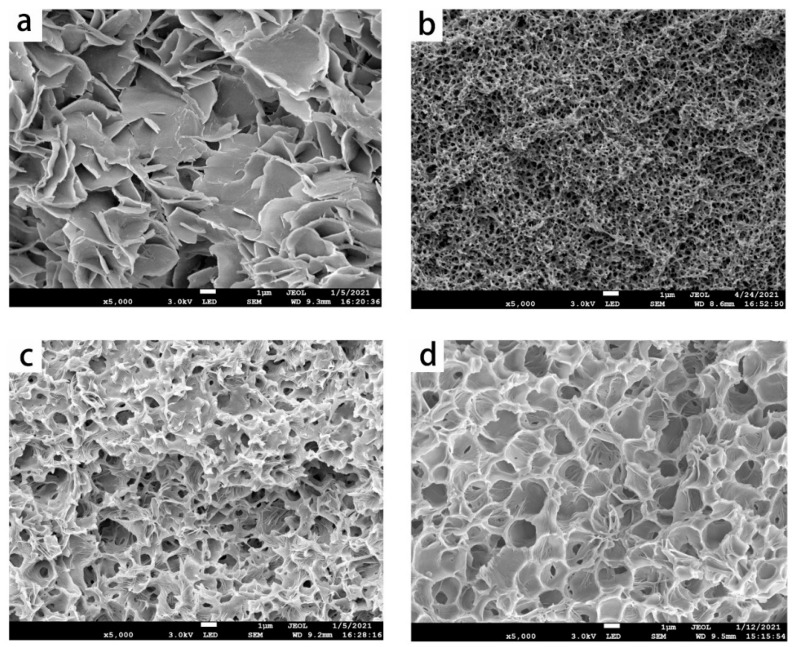
The cross-section structures of the PMP piece membranes prepared by different fatty acids with a PMP concentration of 30 wt%: (**a**–**d**) represent the fatty acid of LA, MA, PA and SA, respectively.

**Figure 3 membranes-11-00622-f003:**
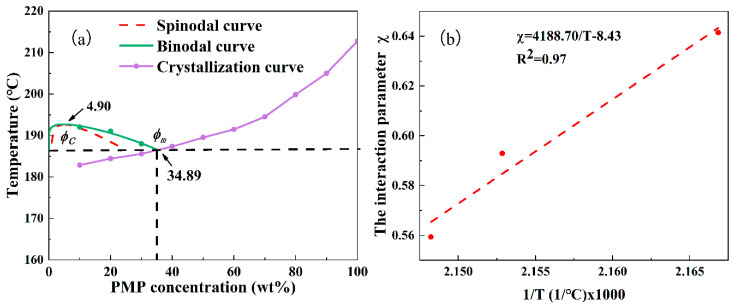
(**a**) The phase diagram of the PMP-MA system, and (**b**) the χ versus 1/T curve of the PMP-MA system.

**Figure 4 membranes-11-00622-f004:**
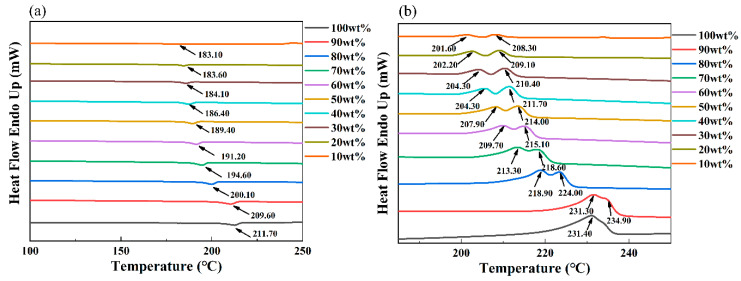
The DSC curves of the PMP-MA system with different PMP concentrations: (**a**,**b**) represent the crystallization curves obtained during the cooling period and the melting curves obtained during the heating period, respectively.

**Figure 5 membranes-11-00622-f005:**
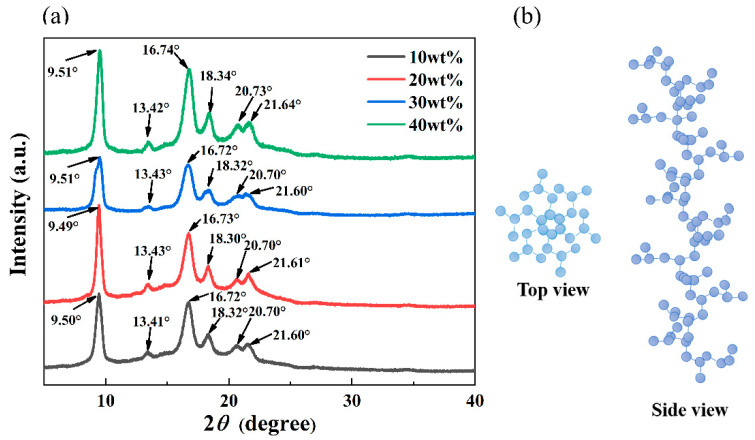
(**a**) the X-ray diffractograms of the PMP-MA system prepared with different PMP concentrations; (**b**) a schematic diagram of the top and side views of the carbon chain conformation in the crystalline phase of PMP [50].

**Figure 6 membranes-11-00622-f006:**
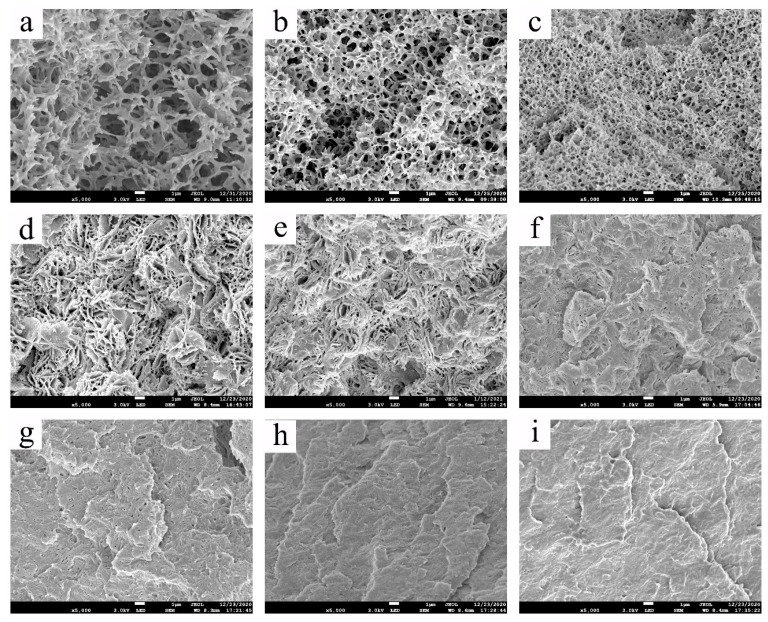
The cross-section structure of the PMP-MA piece membranes with different PMP concentrations: (**a**–**i**) shows 10, 20, 30, 40, 50, 60, 70, 80, and 90 wt%, respectively.

**Figure 7 membranes-11-00622-f007:**
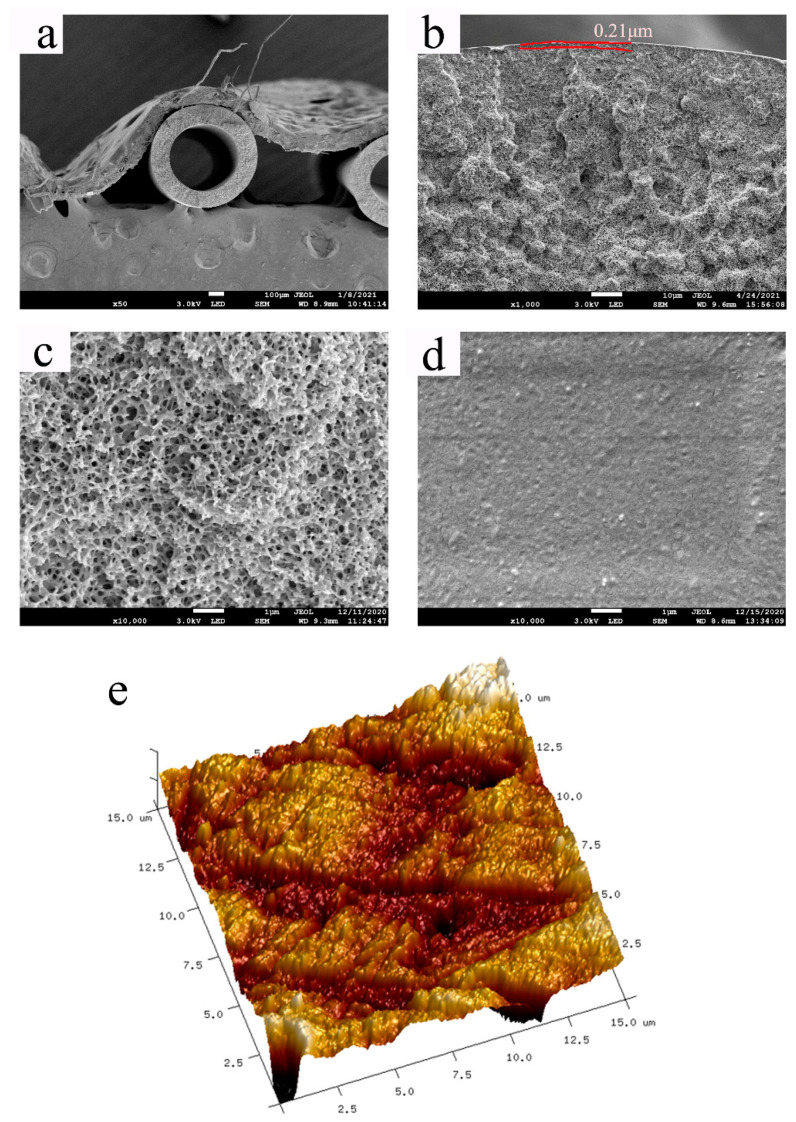
The morphology of the PMP HF membrane: (**a**) the overall cross-section structure, (**b**) the cross-section near the outer surface, (**c**) the partial cross-section structure; (**d**) the outer surface of the membrane; (**e**) the surface roughness of the membrane.

**Table 1 membranes-11-00622-t001:** A summary of the key results of the previous research works related to the PMP membrane preparation via TIPS from 1990 to 2020.

Year	Main Authors [References]	Key Diluents	Membrane Geometry	Comments
1990	Lloyd D.R. [6]	mineral oil (MO) *	Flat sheet (FS)	Membranes prepared presented flaky crystal structures due to the solid-liquid phase separation.
2003	Müller M.O. [8]	dioctyl adipate (DOA) + glycerol triacetate (GTA);	HF	Membranes prepared presented bicontinuous cross-section structures beneath an ultrathin surface layer.
2004, 2007	Kessler E. [9,10,11]	dibutyl phthalate (DBP) *; diphenyl ether (DPE) *; DOA *; Palm kernel oil *; coconut oil *; dioctyl phthalate (DOP) + GTA; DOA + GTA; DBP + GTA	HF	Membranes prepared presented bicontinuous cross-section structures beneath ultrathin surface layers for all the diluent mixtures and some single diluent systems (such as DPE, Palm kernel oil, and coconut oil)
2007–2010	Zhang J. [12,13,14]	dioctyl sebacate (DOS) + dimethyl phthalate (DMP); DBP + DOP; DOP *; DOA *	FS	Spherulitic structures were formed when the DOP or DOA was the single diluent and could be replaced by cellular structure by adding a poor solvent; Membranes prepared presented bicontinuous structures as the weight ratio of DOS:DMP = 1:1
2009	Xia D. [7]	DPE *	FS	Variant pore structures were formed at different PMP concentrations and cooling rates; spinodal decomposition occurred and bicontinuous structure could be found as the PMP concentration was lower than 30 wt%.
2016	Li L. [15,16,17,18]	DOP *; DPE *; DBP *; DBP + DOP;	HF	DOP, as a good diluent, provided smaller and more inter-connected pores, with the highest porosity and gas permeation values; DPE and DBP, as relatively poor diluents, presented larger and less inter-connected pores, with lower porosity and gas permeation values.
2017	Voigt I [19]	DOA + GTA	HF	Membranes prepared presented bicontinuous cross-section structures beneath an ultrathin surface layer.
2019–2020	Jia J. [20,21,22]	dibutyl sebacate (DBS) + castor oil (CO); methyl 12-hydroxystearate (MHS) + DOA; MHS + dimethyl phthalate (DMP);	HF	Membranes prepared presented asymmetric bicontinuous cross-section structures beneath dense surface layers.

Note: The ‘*’ mark represents a single diluent component, and no ‘*’ mark represents a diluent mixture.

**Table 2 membranes-11-00622-t002:** The chemical formulas, the molecular structures, and the basic physical properties of the PMP and fatty acids adopted in the work.

Component Name	ChemicalFormula	Structure	Melting Point/°C	Boiling Point/°C	The Molecular Weight/Da
PMP	[C_6_H_12_]_n_	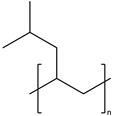	231.40	-	87,200
LA	C_12_H_24_O_2_	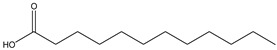	44–46	298.90	200.36
MA	C_14_H_28_O_2_	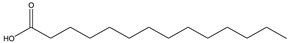	58.00	250.50	228.37
PA	C_16_H_32_O_2_	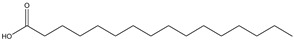	63.10	351.00	256.42
SA	C_18_H_36_O_2_	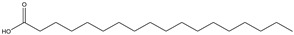	67.00~69.00	361.00	284.48

**Table 3 membranes-11-00622-t003:** The Hansen solubility parameters and the membrane structures of the different PMP-diluents systems.

Polymer or Diluent	$\delta_{D}$ (MPa^1/2^)	$\delta_{P}$ (MPa^1/2^)	$\delta_{H}$ (MPa^1/2^)	Ra	Membrane Cross-Section Structure
PMP	16.60 ^b^	4.80 ^b^	5.20 ^b^	-	-
LA	15.80 ^b^	3.60 ^b^	6.80 ^b^	2.60	Flaky crystal with a PMP concentration of 20–30 wt%.
MA	15.70 ^b^	3.00 ^b^	6.00 ^b^	2.70	Bicontinuous with a PMP concentration of 20–30 wt%.
PA	15.70 ^b^	2.40 ^b^	5.20 ^b^	3.00	Cellular with a PMP concentration of 20–30 wt%.
SA	15.60 ^b^	1.70 ^b^	4.30 ^b^	3.70	Cellular with a PMP concentration of 20-–30 wt%.
DBP	17.80 ^a^	8.60 ^a^	4.10 ^a^	4.60	Bicontinuous with a PMP concentration of 30–35 wt% [17,18].
DPE	19.50 ^a^	3.40 ^a^	5.80 ^a^	6.00	Bicontinuous with a PMP concentration of 30–35 wt% [7].
DOP	16.60 ^a^	7.00 ^a^	3.10 ^a^	3.10	Bicontinuous with a PMP concentration of 30–35 wt% [19].
DOA	16.70 ^a^	2.00 ^a^	5.10 ^a^	2.60	Flaky crystal with a PMP concentration of 20–60 wt% [16].

Note: The superscript ‘^a^’ presents that the values came from the manual of Hansen solubility parameters, and the superscript ‘^b^’ presents that the values were calculated by the group contribution method.

**Table 4 membranes-11-00622-t004:** The melting enthalpy and crystallinity of the PMP-MA system with different PMP concentrations.

PMP Concentration/wt%	Melting Enthalpy/J/g	Initial Crystallization Temperature/°C	Final Crystallization Temperature/ °C	Peak Crystallization Temperature/ °C	Peak Temperature Difference/°C	Crystallinity/%
10	1.27	181.10	185.90	183.10	4.85	10.80
20	3.98	180.10	189.20	183.60	9.07	16.96
30	5.20	179.80	191.00	184.10	11.18	14.76
40	7.23	180.70	192.70	186.40	11.97	15.86
50	9.22	184.30	194.30	189.40	10.02	15.74
60	11.10	185.50	195.50	191.20	10.02	15.79
70	13.40	186.10	199.40	194.60	13.26	16.33
80	17.16	192.50	205.70	200.10	13.24	18.30
90	19.41	199.10	216.00	209.60	16.86	18.40
100	23.56	201.70	218.90	211.70	17.26	20.10

**Table 5 membranes-11-00622-t005:** The configuration information and gas permeability performance of the PMP membranes and the corresponding values of the comparison obtained from the literature of Table 1.

Property	Value	Literature Value
Outer diameter/μm	900 ± 100	1000 ± 200 [16,17]
Inner diameter/μm	600 ± 100	600 ± 200 [16,17]
The thickness of the dense surface layer/μm	0.21	2–3 [17,19]
Porosity/%	60–70	40–70 [17,19]
Surface contact angle/°	105.80	-
Tensile strength/cN	96	40–180 [9]
Nitrogen flux/mL·(bar·cm^2^·min)^−^^1^	8.20 ± 0.10	0–1 [12]

## Data Availability

Not applicable.

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
