# Peer review of "A Novel Green Diluent for the Preparation of Poly(4-methyl-1-pentene) Membranes via a Thermally-Induced Phase Separation Method"

_membranes, 2021, doi:10.3390/membranes11080622_

Round 1
Reviewer 1 Report
Major comments:
The study is focused on A novel green diluent for the preparation of poly (4-methyl-1-pentene) membranes via thermally induced phase separation method. The research is very limited in the aspects of the study and covers only a few points of the whole study. The literature study is very weak and outdated, and there is no proper linkage between different sections of the papers. The language of the paper also needs further improvements as it contains several mistakes. The equations should be written using equation writer instead of the same word format and references should be divided properly and should not be generic. The headings also need changes as they should be based on obtained results instead of the equipment used. The English language used in the manuscript needs improvements, as there are some punctuation and grammatical mistakes throughout the manuscript. Sentences need more clarity and better construction. Some figures need more transparency; special focus is required in labelling the axis and titles. Overall, the paper needs some structural and literature revisions to meet the requirements of the journal. It is obvious the quality of the manuscript does not meet the standards of the membranes Journal in its present form, therefore should be rejected or needs major revisions.
Introduction:
The introduction is very short and too general, therefore need more specific information related to the present research. It should be more focused on the topic of interest. The literature in the introduction needs more updates and the latest literature should be used for the studies. The introduction needs to be more emphasized on the research work with a detailed explanation of the whole process considering past, present, and future scope. The conventional fuels and technologies need to be explained well to indicate the relevance of the research work. It needs to be strengthened in terms of recent research and updated literature review in this area with possible research gaps It is strongly recommended to add a recent literature survey about different types of novel materials, membrances, synthesis techniques that are used in order to construct hybrid membranes system. Research gaps should be highlighted more clearly and future applications of this study should be added.
Specific comments:
- The title is not appropriate, authors are advised to revise the tile, which should be comprehensive and novel. Remove the novelty word, already several researchers published.
- Abstract: It is suggested to add some background with few objectives and possible applications of this study and highlight the novelty of this work clearly. The abstract only contains some parameters without any process conditions or key values from results, which is insufficient to delineate the whole pictures of contribution and possible application of this study.
- Revise keywords add more specific and novel keywords with broader meanings (5-7 words).
- Most of the information presented in the introduction is too general and very common. Therefore, should be omitted and replaced by some latest research and their findings.
- Major formatting changes including proper paragraphing is required throughout the report.
- Page 6, section 2.4 Textural characterizations: Authors have not mentioned what standard methods have been followed, please compare SEM/XRD techniques with following recent studies: Crystal Growth & Design, 2020; 20(4):2406-14.
- The figures and tables numberings must be consistent, you have used Fig. and Figure at different places?
- Most of the headings and subheadings need to be revised in a professional way and for better understating.
- The figure/graphs are very dim and low quality, need to be revised. You can use Origin Lab software if you like.
- Most of the information presented in the discussion is refereed to your own findings/values, please discuss this critically with the literature and other published word.
- The tables/ figures inserted are not explained or discussed well in the text please discuss critically/explain all tables/figures in the text wherever possible.
- The graphs throughout the manuscript are not consistent, some coloured, some black and white with blurry resolution. Revise all graphs with high-quality images and keep all figures as coloured with consistent fonts. The units stated in the graphs axis need to be double-checked.
- Please round off all numbers up to two decimal places throughout the text in the entire manuscript.
- The obtained values in the results are just stated in the text without explaining them. Explain the reasons behind your trends/values and discuss them critically with literature.
- Revise figures in the manuscript. Draw all figures in high-quality figures should be coloured and attractive.
- More recent research about types of novel materials, sustainable membranes development should be added to make the background and discussion more strong: TrAC Trends in Analytical Chemistry, 2020; 132:116066.
- All figures numbering needs to be carefully checked and revised.
- All figures captions should be revised in a more meaningful way.
- Avoid an abundance of references do not cite more than 2 references in a single place. Correct all these types of references throughout the manuscript.
- The conclusions only talk about some studied parameters, which is insufficient to depict the whole picture of the contribution of this study. The authors are advised to write the conclusions in a comprehensive way and should contain key values, suitability of the applied method, the major findings, contributions and possible future outcomes.
- References: The authors are advised to revise this section, including the latest reference. Please see some suggestions in the specific comments and in the ‘introduction’ section.
Reviewer 2 Report
The work "A novel green diluent for the preparation of poly (4-methyl-1-pentene) membranes via thermally induced phase separation method" has interesting and actual subject of the specific research fields.
The structure and general presentation were very good.
The methodology is adequate.
The result are clear, interesting and useful.
The conclusions were based on the obtained results.
The reference covering very well the subject of the research.
Author Response
Reviewer 2:
The work "A novel green diluent for the preparation of poly (4-methyl-1-pentene) membranes via thermally induced phase separation method" has interesting and actual subject of the specific research fields. The structure and general presentation were very good. The methodology is adequate. The results are clear, interesting and useful. The conclusions were based on the obtained results. The reference covering very well the subject of the research.
Response :
Thank you for your appreciation and recognition of this manuscript.
We have has been adjusted the title, abstract, keywords, discussion, and conclusion of the article. All the changes are made to enrich the content of the article and make the gist of this article more prominent.
Reviewer 3 Report
PMP is a semi-crystalline polyolefin and has several interesting properties, such as high thermal stability, good chemical resistance, excellent gas permeability, and biosecurity. It is of practical importance as a membrane-making polymer, and the PMP membranes have been used as the essential membrane materials applied in extracorporeal membrane oxygenation (ECMO) systems, which is an extracorporeal device of providing prolonged cardiac and respiratory support to persons whose heart and lungs are unable to provide an adequate amount of gas exchange or perfusion to sustain life. As the core materials to realize the gas exchange in the ECMO systems, the PMP membranes should possess structures that consist of a uniform and bicontinuous porous cross-section, and a thin dense surface layer to keep a high gas permeability. In this manuscript, the PMP HF membranes were prepared via the TIPS process with a novel green diluent, MA. Firstly, combining the Hansen solubility parameter and the resultant membrane structure, MA was selected to be the optimal diluent among a series of natural green fatty acids. I am pleased to send you moderate comments. The results and theme of this paper is interesting. Generally, this manuscript makes fair impression and my recommendation is that it merits publication in this Journal, after the following major revision:
- The detailed literature review indicates efforts made by the authors. The coherence of the related work, however, is still not clear. It may help the authors by answering the following questions: Why are these works relevant? Which specific problems were addressed? How are the previous results related with the latest work? What are the outstanding, unresolved, research issues? Which of them has been solved by the proposed study? Answering the questions leads to the novelty of the proposed work naturally. Besides, the current one is nothing but a literature review. Why their work is important comparing to previous reports? I think this is essential to keep the interest of the reader.
- The phase diagram of the PMP-MA system showed that the monotectic point was 34.8 wt%, which meant with the increase of the PMP concentration, the phase separation process was gradually changed from the L-L to the S-L phase separation. As a result, the PMP HF membranes were fabricated with a PMP concentration of 30 wt%. The resultant membranes presented that the porosity was 70%, the tensile strength was 96 cN, and the nitrogen flux was 8.2±0.1 mL·(bar·cm2·min)-1. The authors should give some explanation on above conclusions and data.
- In Fig. 4, the authors should give the explanations for the difference of data collected from different sources.
- At present, the PMP membranes applied in the ECMO systems are mainly prepared via thermally induced phase separation (TIPS), since the TIPS process has better controllability in the membrane preparation and excellent membrane performances such as high porosity and good mechanical property can be easily attained. The membranes are found to serve in many practical applications, such as porous materials [International Journal of Heat and Mass Transfer, 2019, 137:365-371], fuel cell [International Journal of Hydrogen Energy, 2018, 43(37):17880-17888], and medical science [1 to 2 references], etc. Authors should introduce some related knowledge to readers. I think this is essential to keep the interest of the reader.
- Experimental part, Although the results look “making sense”, the current form reads like a simple lab report. The authors should dig deeper in the results by presenting some in-depth discussion.
- Please expand the motivation, the problem context, clarify the problem description, and (if possible) add specific objectives.
- Please, expand the conclusions in relation to the specific goals and the future work.
- English grammar and syntax has to be checked carefully throughout the manuscript. There are several grammatical mistakes in the manuscript and it is very difficult to follow anything if they are not corrected.
Round 2
Reviewer 3 Report
The attached manuscript is ok.